# Ecophysiological Features Shape the Distribution of Prophages and CRISPR in Sulfate Reducing Prokaryotes

**DOI:** 10.3390/microorganisms9050931

**Published:** 2021-04-27

**Authors:** Roberto Orellana, Alejandra Arancibia, Leonardo Badilla, Jonathan Acosta, Gabriela Arancibia, Rodrigo Escar, Gustavo Ferrada, Michael Seeger

**Affiliations:** 1Laboratorio de Biología Celular y Ecofisiología Microbiana, Facultad de Ciencias Naturales y Exactas, Universidad de Playa Ancha, Leopoldo Carvallo 270, Valparaíso 2360001, Chile; xrique@gmail.com (A.A.); leonardo.badilla@alumnos.upla.cl (L.B.); 2Departamento de Estadística, Pontificia Universidad Católica de Chile, Avda. Vicuña Mackenna 4860, Santiago 7820436, Chile; jonathan.acosta@mat.uc.cl; 3Laboratorio de Microbiología Molecular y Biotecnología Ambiental, Departamento de Química & Centro de Biotecnología Daniel Alkalay-Lowitt, Universidad Técnica Federico Santa María, Avenida España 1680, Valparaíso 2390123, Chile; gba.arancibia@gmail.com (G.A.); rodrigo.escar.13@sansano.usm.cl (R.E.); gustavo.ferrada@alumnos.usm.cl (G.F.); michael.seeger@usm.cl (M.S.)

**Keywords:** prophages, sulfate-reducing prokaryotes, sulfate reduction, ecophysiology

## Abstract

Sulfate reducing prokaryotes (SRP) are a phylogenetically and physiologically diverse group of microorganisms that use sulfate as an electron acceptor. SRP have long been recognized as key players of the carbon and sulfur cycles, and more recently, they have been identified to play a relevant role as part of syntrophic and symbiotic relations and the human microbiome. Despite their environmental relevance, there is a poor understanding about the prevalence of prophages and CRISPR arrays and how their distribution and dynamic affect the ecological role of SRP. We addressed this question by analyzing the results of a comprehensive survey of prophages and CRISPR in a total of 91 genomes of SRP with several genotypic, phenotypic, and physiological traits, including genome size, cell volume, minimum doubling time, cell wall, and habitat, among others. Our analysis discovered 81 prophages in 51 strains, representing the 56% of the total evaluated strains. Prophages are non-uniformly distributed across the SRP phylogeny, where prophage-rich lineages belonged to Desulfovibrionaceae and Peptococcaceae. Furthermore, our study found 160 CRISPR arrays in 71 SRP, which is more abundant and widely spread than previously expected. Although there is no correlation between presence and abundance of prophages and CRISPR arrays at the strain level, our analysis showed that there is a directly proportional relation between cellular volumes and number of prophages per cell. This result suggests that there is an additional selective pressure for strains with smaller cells to get rid of foreign DNA, such as prophages, but not CRISPR, due to less availability of cellular resources. Analysis of the prophage genes encoding viral structural proteins reported that 44% of SRP prophages are classified as Myoviridae, and comparative analysis showed high level of homology, but not synteny, among prophages belonging to the Family Desulfovibrionaceae. We further recovered viral-like particles and structures that resemble outer membrane vesicles from *D. vulgaris* str. Hildenborough. The results of this study improved the current understanding of dynamic interactions between prophages and CRISPR with their hosts in both cultured and hitherto-uncultured SRP strains, and how their distribution affects the microbial community dynamics in several sulfidogenic natural and engineered environments.

## 1. Introduction

For many years, bacteriophages have been considered the dark matter of the biological world due to their high abundance, ubiquitousness, dynamic population, genetic diversity, and indecipherable mosaic genomic architecture [1,2]. The term bacteriophage refers to viruses that are capable to infect bacteria and archaea. It comprises temperate, non-temperate phages, and defective prophages. Temperate phages are bacteriophages that are genetically capable of exhibit both, lysogenic, and productive cycles. The lysogenic state is characterized by the incorporation of the viral genome into the host chromosome to become prophages, where viral gene expression remains at a minimum level [3,4,5]. A fraction of prophages surreptitiously persists in the genome until they meet one of the two possible fates. The first one is the activation by chemical or physical changes or spontaneous induction leading the excision of the viral genome and further production and release of new viral progeny [6]. Those events produce the lysis of the prokaryotic host cells, releasing dissolved organic matter that can be mineralized in the environment. The second possible fate is the loss of their ability to induce as a result of the gradual elimination or genomic streamlining, in which genes that comprise the prophages are slowly removed [7]. The resulted prophage elements remain as defective prophages, prophage remnants, satellite viruses, and isolated phage genes, which are often present in microbial genomes. Non-temperate phages are viruses that are unable to display lysogenic cycles [4].

The prophage–host interaction can have positive or negative effects for both the host and the virus, which depends on the environment. On one hand, carrying integrated phages that kill the lysogenic cell after eventual induction is considered an extreme relationship for the host, but it is convenient way to silently replicate inside a protective cellular environment for the virus. Moreover, replication and maintenance of extra bacterial DNA is a metabolic burden that affects the long-term fitness of the host. On the other hand, carrying prophages can result on several advantages. For example, it has been shown that prophages can protect host against further viral infections, and phagocytosis [3,6,8], as well as play a role in the osmotic, oxidative, and acid stress responses [9]. Several phages encode auxiliary metabolic genes (AMGs), a collection of genes that encodes metabolic functions that are additional to the critical processes for phage replication [10]. Genes encoding proteins involved in photosynthesis, carbon metabolism, stress tolerance and nucleic acid synthesis are examples of AMGs that has been characterized in marine cyanoviruses [11,12].

Sulfate reducing prokaryotes (SRP) are a phylogenetically, physiologically, and nutritionally diverse group of microorganisms that have in common the use of sulfate as an electron acceptor, which results in the production of sulfide as the end-product [13,14]. The majority of SRP are substrate-versatile and capable to use several fermentation products as electron donors, including hydrogen and organic compounds such as acetate, ethanol, formate, lactate, pyruvate, malate, and succinate [15,16]. In addition to the sulfur cycle, SRP play a key role in the carbon and mercury cycles in many anoxic environments where sulfate is available. SRP metabolize around 50% of organic carbon in marine sediments [17], and support a high fraction of anaerobic organic matter degradation in wetlands and freshwater lakes [18,19].

Although their environmental relevance, the prevalence of phages and how their distribution and dynamic could affect the ecological role of SRPs in global biogeochemical cycles have been scarcely studied. At the level of pure cultures, traditional recovery approaches using mitomycin C or UV radiation has provided evidence that phages are capable to infect the model strain *Desulfovibrio vulgaris* str. Hildenborough [20,21,22]. Further reports on co-culture of phages extracted from *D. vulgaris* str. Hildenborough and its close relative strain DePue, shed light on the occurrence of cross-infection within SRP and also how prophages become a relevant source of genomic sequence divergence [23,24]. The incorporation of additional methods based on filtration, estimation of bacterial density, concentration or enrichment of virions by centrifugation have improved the recovery of phages that infect SRP, including recoveries from *Desulfovibrio salexigens* and *Desulfovibrio aespoeensis* [25,26]. These studies have been very important to describe lytic phages in SRP, however, they have been restricted to few strains, due to the many technical restrictions and particularities that are inherent in the culturing of SRP.

The combination of culturable-based methods with analysis using next-generation sequencing (NGS) technologies allows to broaden our understanding of phage ecology in environments under sulfate reduction [27]. Several methods for detection of phages based on searching phage-related sequences in microbial genomes have been used in a broad range of microbial species [28,29,30]. However, there is no uniform criteria for this kind of survey, making it difficult to discriminate between inducible prophages, prophages showing deletions or insertions, and prophage remnants [7]. Despite those technical limitations, a recent analysis of the 47 genomes belonging to the genus *Desulfovibrio* highlighted the prevalence of phages on the SRP community; however, additional phages may remain to be discovered and key environmental drivers that governs phage–SRP interaction and lysogeny should be determined [31].

In this study, we conducted a comprehensive survey of prophages and CRISPR arrays in a total of 91 publicly available genomes of SRP, integrating those results with several genotypic, phenotypic and physiological traits, including genome size, cell volume, minimum doubling time, cell wall, presence of flagella, and habitat. These results will contribute to improve the understanding of the abundance and distribution of prophages and CRISPR and how they affect the microbial community dynamics in several sulfidogenic natural and engineered environments, including anoxic sediments, muds, stream and rivers, and sewage digesters.

## 2. Material and Methods

### 2.1. Searching for Prophages and CRISPR Elements in the Genomes of SRP

The search of prophage-like sequences in publicly available SRP genomes was conducted through a combination of two automatized approaches that have different strategies. Initially, the SRP genomes were checked by PHAge Search Tool Enhanced Release (PHASTER) web server (http://phaster.ca/ (accessed on 1 February 2020)) using default parameters for closed and WGS data [30]. PHASTER bases the detection of prophage-like regions enriched with protein-coding phage homologs, and it has been previously tested in SRP [31]. In addition, an independent strategy was based on PhiSpy, that search prophage-like elements by ranking genomic regions enriched in typical viral features, such as phage insertion points, protein length, AT and GC skew, among others [32]. PhiSpy was used locally, utilizing the training set for the analysis of each SRP as its closest relative on basis of the phylogenetic analysis of 16S rRNA gene sequences (Appendix A). Although few prophages have the ability to remain as episomal or plasmidial [33,34,35], we performed both surveys in the extrachromosomal elements in the 13 SRP that contain closed genomes and reported plasmids (Appendix A). Eight out of 13 plasmids were too short to be analyzed by PhiSpy, therefore, in those cases both chromosomes and plasmids were concatenated and analyzed together. Afterwards, we selected the overlapping prophage-like elements between both independent approaches, PHASTER and PhiSpy, and constructed the list of genomic regions predicted as prophages (Appendix A). The distribution of prophages was associated to SRP phylogeny. For that purpose, a DsrAB database was reconstructed integrating sequences from previous reports with new ones. The last sequences were manually retrieved from NCBI aminoacid databases using the following terms “DsrAB”, “DsrA”, “DsrB”, and also ‘dissimilatory (bi) sulfite reductase’, ‘dissimilatory sulfite reductase’, ‘dissimilatory sulfite reductase’. Afterwards, a neighbor-joining phylogeny of the DsrAB dissimilatory (bi)sulfite reductases was reconstructed including 70 sequences out of 91 SRP analyzed. The alignment was made by MUSCLE algorithm [36] and manually curated in Geneious [37] to a final alignment of 523 unambiguously aligned positions. Resampling was made by bootstrapping method using 100 resampling and support threshold was set at 50%. DsrAB phylogeny has the advantage to cover the extensive SRP diversity identifying differences between those strains that utilize an oxidative versus reductive sulfur metabolism. However, one potential limitation of DsrAB-based phylogeny resides in the fact that members of the major SRP lineages are widely distributed among very different environments. In consequence, we made a classification of lysogenic SRP mainly grouped in three different clusters, including two clusters that were abundant in *Desulfovibrio* strains and a third cluster dominated by Firmicutes.

The prediction of the protein-coding genes of the prophages was made by prodigal with the following command: ./prodigal –i fasta.nucleotide –o my.genes –a proteins.faa using nucleotide sequences of the prophages (Data Set 1). For those prophages that were shorter than 20,000 bp, the command was modified by adding the flag –p meta, as recommended for short sequences, including plasmids, phages, and viruses [38]. The outputs of prodigal that include amino acid sequences are available in the Data Set 2. Prophage annotation was made by two different approaches. The first approach was based on prediction of gene functions using DIAMOND based on RefSeq97 and it was applied in the annotation of all prophages found in this study [39] (Data Sets 3 and 4). A second approach was based on manual inspection and the complementation of the former annotation with annotation made with DNA Master (http://cobamide2.bio.pitt.edu/computer.htm (accessed on 4 September 2020)). This annotation was completed with prophages found in *D. vulgaris* str. Hildenborough, *D. vulgaris* DP4, *Desulfocurvibacter africanus* DSM 2603, *Desulfovibrio vulgaris* str. Miyazaki, *Desulfotomaculum hydrothermale* Lam5, *Desulfotomaculum ruminis* DSM 2154, and *Desulfotomaculum reducens* MI-1. Those results were classified in the following two categories: high confidence level for those genes with *e*-value less than 10^−4^ and alignment length longer than 100, and low confidence level for those genes with *e*-value more than 10^−4^ and alignment length shorter than 100 (Data Set 5). Functional annotation of prophages was manually checked, and we searched for the words “terminase”, “integrase”, “nuclease”, “transposase”, “DNA primase”, and “lysin” in the protein coding features as markers of functions related to the integration or excision of phages. We also manually checked the words “portal”, “head”, “capsid”, “tail”, “coat”, and “baseplate”, as key elements of the viral structures that remained in prophages. Structural genes were also verified using VIRFAM [40]. We performed sequence comparison among all of the open reading frames (ORFs) of SRP prophages by DIAMOND, an alternative for BLASTX [39], and homologs were defined as proteins with 30% of similarity and *e*-value of 10^−4^. Homologies were represented with graphs created by the Networkx Python library [41] using their available Graphviz interface for generating the graph layout, in combination with Pandas to manipulate and filtering the blast tables into Networkx graph data structure and Matplotlib for plot customization. Synteny across different prophages was represented by Dotplots made with Gepard [42], and Circos Sofware [43].

We searched Clustered Regularly Interspaced Short Palindromic Repeats (CRISPR) locus in the genome of SRP by CRISPRCasFinder (https://crisprcas.i2bc.paris-saclay.fr/CrisprCasFinder/Index (accessed on 5 August 2020)) using the default variables and also allowing the alternative detection of truncated repeats [44]. We considered a CRISPR array was found when the evidence level ranged between 2 and 4, which is the most stringent criteria considered to discriminate them from spurious CRISPR-like elements. We also checked our results with predictions of CRISPR arrays by using CRISPRFinder (https://crispr.i2bc.paris-saclay.fr/Server/ (accessed on 30 August 2020)) that was run locally using default parameters [45]. CRISPR in extrachromosomal elements were also examined.

### 2.2. Ecophysiological Analysis of SRP

An extensive meta-analysis of morphological, physiological and metabolic features and lifestyles of the SRP was performed based on the information for the 91 individual species derived from literature. This analysis included growth conditions, such as pH ranges and temperature ranges (with the terms “mesophiles”, “thermophiles”, and “psychrotrophs” used as descriptors); cell morphology aspects, such as Gram staining properties of the cell wall (“Gram-positive” or “Gram-negative”), motility (presence or absence of flagella), cell shape (bacillus, coccus, coccobacillus, and others); metabolic traits, such as electron acceptors and electron donors for growth (with the terms “complete” or “incomplete” oxidizer); and growth characteristics, such as the doubling time for optimum growth that was estimated from research that included culturing of pure strains. According with those estimations, SRP were grouped in two categories based on the capability to duplicate (under optimum conditions) in ≤12 h or >12 h, as fast-growing or slow-growing strains, respectively. In addition, the analysis included the source of isolation for each strain, or the habitat where material that allowed the reconstruction of the genomes was obtained, as in the case of Candidatus *Desulforudis audaxviator* MP104C, *Desulfatitalea* sp. BRH_c12, and *Desulfobulbaceae* bacterium BRH_c16a. The habitats were classified into the following categories: “freshwater”, “brackish water”, and “marine sediments”, “soil”, “impacted or engineered system”, and “animal or plant-associated”. Furthermore, cell volume (μm^3^) data was recovered from previous research and, for those strains for which there was no prior information, this was estimated by assessing the dimension of cells in transmission electron micrographs. To estimate the volume of rods, length and width of cells were averaged, and then we applied the formula for the volume of a cylinder (V = π * r^2^ * L). For ovoid shaped cells, we applied the formula for the volume of an ellipsoid (V = 1.333 * (π * a * b * c)), considering the average of half-cell length (a) and width (b and c). Finally, for cocci, we based the estimates on the formula for the volume of a sphere (V = 1.333*(π)*(r^3^)). After calculations of all SRP for which data was available, they were classified as small (<1 μm^3^), middle-size (1–2.5 μm^3^), and big cells (>2.5 μm^3^). A second parameter estimates was surface cell area (um^2^). It was estimated using the area of a cylinder (A = 2 * π * r^2^ * π * r * L) for rods, the area of the sphere for cocci (A = 4 * π * r^2^), and approximated area of an ellipsoid for ovoid shaped cells (A ≈ 4π [(a^p^ * b^p^) + (a^p^ * c^p^) + (b^p^ * c^p^)/3]^1/p^), where p is equal to 1.6075. SRP were classified as small surface area (<6 μm^3^), middle surface area (6–12 μm^3^), and large surface area (>12 μm^3^).

### 2.3. Statistical Analysis

The statistical techniques used in this paper are Principal Component Analysis (PCA), *t*-test for difference of means, and Wilcoxon test. PCA was used to gain some understanding of how ecophysiological traits may affect lysogeny and the prevalence of CRISPR elements in the SRP community. PCA is a technique used in dimension reduction when there are many variables. Commonly, a few dimensions can explain 80% or more of the variability of the data. In addition, the graphical visualization of the first two components provides a suitable tool to detect hidden/unknown patterns among several variables [46,47]. The *t*-test and Wilcoxon tests, and parametric and non-parametric techniques, respectively, are tools to assess the difference between two variables. In these tests, the null hypothesis is that the centers of both variables are equal. For the *t*-test, the difference between two variables is measured through the difference in averages of each variable, which is standardized by the standard deviation, while the Wilcoxon test is based on the location-shift, meaning that the distribution of the two variables is the same except for a constant. Both tests have advantages and disadvantages, as well as their assumptions, and therefore, their use should be adjusted appropriately to each aim. The *t*-test assumes independence between each observation and the normality of both variables. The Wilcoxon test assumes independence between each observation and identical distribution in both populations, but not necessarily normal distribution. The principal advantage of the *t*-test, over the Wilcoxon test, is their higher power statistic (reject when you have to reject) but is very poorly robust in the presence of outliers. The principal advantage of the Wilcoxon test is its robustness in presence of outliers [48,49,50,51]. All analysis were conducted in the software R [52].

### 2.4. Experimental Analysis

*Desulfovibrio vulgaris* str. Hildenborough (DSM 644), *Desulfocurvibacter africanus* 2603 (DSM 2603), *Desulfovibrio desufuricans* subsp. *aestuarii* (DSM 10141) were obtained from the DSMZ, and cultured at 30 °C under strict anaerobic conditions, as previously described [53]. For preparing for prophage induction experiments, strains kept at −80 °C (glycerol 15%) were cultured in liquid media. *D. vulgaris* str. Hildenborough and *Desulfocurvibacter africanus* 2603 were cultured in LS4D medium [54], whereas *Desulfovibrio desulfuricans* subsp. *aestuarii* was cultured in 195-C medium with strain specific modifications. We evaluated the possible induction of phages using increasing amounts of mitomycin added in a final concentration of 1 to 5 micrograms mL^−1^ (Appendix A). The presence of viral-like particles was observed in the microscope after samples were vacuum filtered (Whatman Anodisc inorganic filter membrane, pore size 0.02 μm) and stained with SYBR Gold. These studies were performed with mitomycin added at early exponential phase as well as at early stationary phase, not detecting significant differences. Additionally, 1 mL of filtrates of bacterial cultures induced with mitomycin were added in 10 mL liquid cultures of same strains in both growth stages, finding results like the previous ones (Appendix A). Double layer experiments were performed using the same medium, but adding 0.1 × of iron in order to decrease chances of precipitation upon contact with the H_2_S produced.

## 3. Results and Discussion

In order to evaluate the prevalence of prophage elements in SRPs, in this study a comprehensive survey in 91 publicly available SRP genomes was conducted. The analysis using PhiSpy and PHASTER softwares resulted in the discovery of 81 prophage elements in 51 SRP strains, representing the 56% of the evaluated strains (Appendix A). Thirty-four SRP strains possess a single prophage in their genome, while 17 strains carry two or more prophages (Appendix A). SRP prophages possess an average genome size of 56.12 kb, which is slightly higher compared to the average size of dsDNA temperate phages reported in a broader study [55]. The length of prophages ranges from 13,123 to 169,593 bases in *Desulfatitalea* sp. BRH_c12 and *Desulfosporosinus acidiphilus* SJ4, respectively (Appendix A). Twelve prophage elements (15%) were classified as small prophages (<30 kb). Six were classified as either Myoviridae (5) or Siphoviridae (1), for which their minimum genome size reported in viral web resource ViralZone is 100 kb and 22 kb, respectively (http://viralzone.expasy.org/ (accessed on 11 March 2020)). The other six could not be taxonomically assigned. Furthermore, 75% of those small prophages were present in strains carrying a single prophage, therefore, probably a high fraction of those small prophages are either mobile genetic elements or prophage remnants that may represent ancient insertion events that have lost their ability to induce [6]. For those SRP that have been shown to contain plasmids (13 out of 40 closed genomes), no prophages were found associated to the extrachromosomal elements (Appendix A). The total genome size of the prophages ranged between 13,123 bp and 360,426 bp, and it contributes 0.22–9.55% of the SRP genome.

Prophages are non-uniformly distributed across the SRP phylogeny, where few SRP clusters concentrate a high number of prophages (Figure 1). While prophage-rich lineages belonged to Desulfovibrionaceae and Peptococcaceae, there were no prophages found in strains belonging to the archaea Archaeoglobaceae, and the bacteria Desulfonatronaceae and Nitrospiraceae (Figure 1). The order Desulfovibrionales gather 46 prophages (57% of the total), and the family Desulfovibrionaceae contains 42 prophages (Figure 2). The order Desulfobacterales contains 16 prophages (20% of the total); 10 and 6 of these prophages belong to strains affiliated to the Desulfobacteraceae and Desulfobulbaceae families, respectively. Peptococcaceae (Order Clostridiales) was the only family for which all the strains possess lysogenic prophages (Figure 2). Peptococcaceae strains contain an average of 2.16 prophages per strain and include *Desulfotomaculum acetoxidans* DSM 771 and *Desulfotomaculum reducens* MI-1 that possess more than three prophages. In addition to the high prevalence, prophages that reside in Peptococcaceae and Desulfobulbaceae strains contain significantly larger prophages in comparison to the other SRP families (*p*-value = 0.0089 of Wilcoxon test).

The prophages associated to *Desulfovibrio* strains are grouped in two clusters according to the phylogeny. Cluster number one included two groups of *Desulfovibrio* strains. The first group includes a stable monophylethic set of 13 strains together with five *Desulfovibrio* strains (*D. aespoeensis* Aspo-2, *D. piezophilus* C1TLV30, *D. desulfuricans* ND132, *D. oxyclinae* DSM 11498, and *D. longus* DSM 6739) that branched more independently but have been isolated from more similar environments of strains belonging to cluster 1 than the environment of strains that belong to the cluster 2. In cluster 1, prophages are present in 13 out of 18 strains covering a total of 29 prophages (Figure 1). This cluster includes seven strains, such as *D. africanus* DSM 2603, *D. vulgaris* str. Miyazaki, *D. vulgaris* str. Hildenborough, *D. vulgaris* DP4, *D. alaskensis* G20, *D. fructosivorans* JJ, and *D. aespoensis* Aspo-2, which contain two or more prophages. Furthermore, six SRP carry a single prophage per genome, *Desulfovibrio desulfuricans* ATCC 27774, *D. piger* ATCC 29098, *Desulfovibrio* sp. A2, *D. cuneatus* DSM 11391, *D. alcoholivorans* DSM 5433, and *D. desulfuricans* ND 132 (Figure 1). Within cluster 1, *D. vulgaris* str. Hildenborough has the highest number of prophages in a single SRP, gathering a total of six prophages that occupy the 9.55% of the genome (Appendix A), in accordance with previous reports [31]. Cluster 1 also includes *D. aespoensis* Aspo-2, *D. africanus* DSM 2603, *D. vulgaris* DP4, *D. alaskensis* G20, and *D. fructosivorans* JJ that contain three prophages. These poly-lysogens have been isolated from a wide range of pristine marine, brackish and freshwater sediments and soils, with the exception of *D. alaskensis* G20 that was isolated from production fluids of offshore oilfields in Alaska [57]. A common feature across all those poly-lysogenic SRP is that they are fast-growing motile bacteria and incomplete oxidizers, with the ability to metabolize a huge variety of substrates, including hydrogen (Appendix A). Those capabilities may help them to outcompete other specialized SRP in areas where a diversity of energy and carbon sources are more available, suggesting that their ecological niche is relatively wider than the one for complete oxidizers.

The cluster number 2 included a more diverse set of *Desulfovibrio* strains that grouped well with *Desulfocurvibacter* and *Desulfocurvus* and with strains of the Desulfobulbaceae family besides those belonging *Desulfobulbus* cluster. The distribution of prophages in the cluster 2 is different compared to cluster 1. In cluster 2, nine strains possess a single prophage, whereas only four poly-lysogenic strains are observed. These strains include *Desulfotalea psychrophila* LSv54, *D. africanus* subsp. *africanus* str. *Walvis Bay*, *D. inopinatus* DSM 10,711 and *Desulfobacca acetoxidans* DSM 11109, which have been isolated from a narrow diversity of environments. While the first three strains were isolated from marine sediments off the coast of Svalbard (Norway), Walvis Bay (South Africa), and Venice (Italy), respectively, *D. acetoxidans* DSM 11,109 was isolated from a UASB sludge reactor [58,59,60,61]. In contrast to cluster 1, three out of the four poly-lysogens of the cluster 2 are classified as slow growing bacteria. The single lysogen hosts of cluster 2 include *D. salexigens*, *D. bastinii* DSM 16055, *D. putealis* DSM 16056, *Desulfocurvus vexinensi*s DSM 17965, *Desulfomicrobium baculatum* DSM 4028, *Desulfovibrio* sp. X2, *Desulfurivibrio alkaliphilus* AHT 2, and *Desulfococcus oleovorans* Hxd3. Most of these SRP are incomplete oxidizers, slow growers and have flagella (Appendix A).

The cluster number 3 is restricted to strains belonging to Thermodesulfobacteria, Syntrophobactaraceae, Desulfobulbaceae that belong to the *Desulfobulbus* cluster and Firmicutes (*Desulfotomaculum*). This cluster comprises lysogens belonging to the genus *Desulfotomaculum* (Family Peptococcaceae), including the poly-lysogenic *D. acetoxidans* DSM 771, *D. hydrothermale* Lam5, *D. reducens* MI-1, and *D. ruminis* DSM 2154, which have additional 11 prophages (Figure 1). Cluster number 3 also includes SRP belonging to Desulfobacterales, Thermodesulfobacteriales, Syntrophobacterales, and Desulfarculales order, and except for *Desulfobulbus propionicus* type strain (1pr3), all other strains carry only one prophage per genome. Members of cluster 3 have a broader range of ecophysiological traits compared with clusters 1 and 2. The cluster includes two thermophiles, *Thermodesulfobacterium commune* DSM 2178, and *Desulfotomaculum hydrothermale* Lam5, both incomplete oxidizers isolated from volcanic material from Yellowstone and a hot spring in North-east Tunisia, respectively [62,63]. Lytic phages have been previously characterized from those thermophilic environments, and also from other extreme environments where viral component of microbial community was previously unexpected [64]. Furthermore, two poly-lysogenic strains encompassed in this cluster *D. reducens* MI-1 (3) and *D. hydrothermale* Lam5 (2) have cell wall structure typical of the Gram-positive bacteria [63,65]. Cluster 3 also includes four lysogenic SRP that are slow growers and three strains capable of the complete oxidation of derivatives of organic matter (Appendix A).

Two lysogens belonging to cluster 3 are capable of degrading propionate. One of those strains is *Desulfobulbus propionicus* type strain (1pr3) that possess flexible degradative machinery to metabolize different substrates. Indeed, *D. propionicus* type strain (1pr3) is capable to oxidize propionate to acetate without dependence of syntrophic interactions [66], and also it is capable of producing propionate and acetate as fermentation products when pyruvate or lactate is present in the absence of external electron acceptors [67]. The second strain is *Syntrophobacter fumaroxidans* MPOB that is capable to grow on propionate axenically or syntrophically with other microorganisms, in a process that involves the efficient scavenging of products to cope with the thermodynamic constraint of the propionate oxidation to acetate [68,69,70]. *S. fumaroxidans* MPOB partners with methanogens, such as *Methanospirillum hungatei*, *Methanobacterium formicicum,* or with other sulfate reducing bacteria such as *Desulfovibrio desulfuricans* strain G11. In these syntrophic associations, *S. fumaroxidans* MPOB plays a role of the hydrogen or formate scavenger to keep them at a concentration that makes propionate degradation to be energetically feasible [70]. Interestingly, in our study an alignment of the prophage from *S. fumaroxidans* MPOB (prophage 83R1) against the genome of *D. desulfuricans* strain G11 using tBLASTn showed that 17 out of the 55 ORFs predicted for the prophage are homologs of genes from *D. desulfuricans* strain G11, including genes encoding for the minor capsid protein, major head protein, endonuclease, and phage protease (Data Set 6). It could be interesting to couple the metabolic characterization with in-depth ecological characterization of syntrophic partnerships, which could open opportunities to gain insights into the occurrence and distribution of prophages shared between syntrophs, and how those mechanisms may help them to improve the nutritional interplay in the environment. These observations suggest that, in addition to the broad metabolic flexibility present in lysogenic SRP of cluster 1 and 2, the lysogens of cluster 3 possess physiological flexibility, allowing them to thrive under a wide range of temperatures, salt concentrations, pH, life-styles, and also cope with a broad range of toxic compounds.

### 3.1. CRISPR–Cas Systems Are Present in a High Proportion of SRP

Clustered regularly interspaced short palindromic repeats (CRISPR) is a sophisticated adaptive immune defense system that allow the identification and cleavage of allochthonous DNA, storing the sequences of the invader in series of spacers separated by repeats that permits the host to quickly identify and react upon re-exposure [71]. In contrast to prophages, the presence of CRISPR-Cas arrays constitutes a barrier for the different forms of horizontal gene transfer, constraining transduction [71], natural transformation [72], and conjugation [73]. The CRISPRCas-Finder [44] was applied to search on the analyzed SRPs, since CRISPR represents a valuable tool for defense against phage infections.

The incidence and distribution of CRISPR-Cas systems among a broad range of prokaryotes has been reported. For example, a previous survey indicates that 54% of strains belonging to four *Klebsiella* species contain CRISPR-Cas systems [74]. The presence of this CRISPR was also evaluated in 228 strains belonging to 31 genera of the family Enterobacteriaceae, and 57% of those genomes encode CRISPR arrays [75]. A broader study reports ~50% of prevalence of CRISPR elements in 2110 bacterial genomes [55]. The results of our study showed that 78% (71 out of 91) of SRPs contain at least one CRISPR array, which is considered high in comparison to other prokaryotes. Thirty strains carried a single CRISPR array in their genome, while 41 strains carried more than one CRISPR array (Appendix A). The SRPs surveyed in this study carry a total of 160 CRISPR arrays; at least two of those are present in plasmids (Appendix A). These results suggest that CRISPR systems can be acquired by horizontal gene transfer within sulfidogenic communities [76,77]. The number of CRISPR arrays per genome ranges from 1 to 11 and like prophages, they were non-uniformly spread across the SRP phylogeny displaying a positively skewed distribution towards certain lineages, including Peptococcaceae and Desulfobacteraceae. Two poly-lysogens of the Peptococcaceae, *D. acetoxidans* DSM 771 and *D. ruminis* DSM 2154, were found to contain the higher quantity of CRISPR arrays with a total of 11 and 10, respectively (Figure 1). Except for *Desulfobulbus japonicus* DSM 18378, all the strains belonging to the cluster 3 contain at least on CRISPR array. Even though members of family Desulfovibrionaceae are the contributors to the largest number of CRISPR systems, 36% (13 out of 36) of those species lack CRISPR arrangements, suggesting that other mechanisms may be reinforce for defense against virus in SRP belonging to Desulfovibrionaceae (Figure 2).

Despite our results show there is no significant Pearson correlation between presence and abundance of prophages and CRISPR arrays at the strain level (*p*-value = 0.0727), the distribution of the total number of prophages and CRISPR arrays followed a similar order at the level of Family (Spearman’s Correlation = 0.729; *p*-value = 0.0031) (Figure 2). Meanwhile 56% of the SRP are lysogens, a much higher fraction (76%) of them carries CRISPR arrays, which suggests that there is less evolutionary selection pressure to remove CRISPR arrays than prophages. This imbalance contrast with previous results that found a close to 1:1 ratio between both elements, based on a broader analysis of 7085 prokaryotic genomes [78]. These results can be attributed to strains belonging to the Family Desulfovibrionaceae (Appendix A), for which nine of the strains, *Desulfovibrio vulgaris* str. Hildenborough, *D. africanus* DSM 2603, *D. africanus subsp. africanus* str. Walvis Bay, *D. aespoeensis*, *D. cuneatus* DSM 11391, *D. salexigens*, *D. piezophilus* C1TLV30, *D. putealis* DSM 16,056, and *D. vulgaris* DP4 are lysogenic, however we did not detect the presence of any CRISPR array.

In agreement to prior studies based on a wider set of data [55], our data suggests that SRP lysogens are more likely to carry CRISPR systems than non-lysogenic strains (Odds ratio = 2.15). However, this probability is not significant (95 percent confidence interval range between 0.707–6.91), implying that the difference may be due to sample size rather than a systematic component. In addition, lysogens encode CRISPR arrays that contain a median of 30 spacers, while non-lysogenic SRP contain CRISPR with 25 spacers (Appendix A), both amounts of spacers at a level of memory that could allow an adequate immune response [79]. Surprisingly, four out of the 81 prophages (5%) encode for CRISPR arrays (Appendix A). Although it would seem counterintuitive for prophages to contain typical elements belonging to the host defense against exogenous nucleic acids, it has been previously shown that this is a common feature in several mobile genetic elements, including prophages. For example, CRISPR-Cas arrays were reported as part of mobile genetic elements in diverse bacteria, such as, *Yersinia pestis* and *Vibrio vulnificus* [80]. Indeed, *cas* genes were previously reported as present in the megaplasmid of *D. vulgaris* Hildeborough, but a later report as well as our findings did not detect any CRISPR array in the plasmid [31,76]. Furthermore, the diversity of the SRP CRISPR repertoire and its dynamic interaction with phages and other mobile genetic elements remain to be examined in further detail.

### 3.2. Ecophysiological Traits Shape the Distribution of Prophages and CRISPR in SRP

So far, our results suggested that the trade-off between the energetic burden of maintaining prophages and CRISPR loci versus their contribution to the fitness of the host may be driven by a balance of physiological and environmental factors. Furthermore, we hypothesized that several ecophysiological traits are important to shape the distribution of prophages and CRISPR in SRP. This hypothesis was addressed applying a species-specific approach, retrieving and analyzing several phenotypical and physiological traits from both environmental and experimental studies of the 91 SRP included in the study. These data include type of cell wall (based on Gram stain), motility (presence of flagella), optimum pH and T° for growth, minimum doubling time, cell volume, surface cell area, morphology type, and habitat, based on the environment where strains were either found or isolated (Appendix A).

Principal Component Analysis (PCA) of the eight variables of lysogenic SRP strains showed the first component (PC1) accounted for 25.8% of the total variation in the dataset, and it is highly influenced by genome size, temperature of growth, and cellular volume. Therefore, lysogenic SRP located in the left side contain bigger genomic size and cell volume and often reside in mesophilic environments, in contrast to those lysogenic SRP of the right side of the PCA (Figure 3). The second component (PC2) accounted for 19.9% of the total variation in the dataset, and it is highly influenced by GC content and pH. Despite of the counterintuitive expectation that GC content may be higher in acidic environments, we observed that genomic GC content of lysogenic SRP increases with the optimum pH for growth (Figure 3). A similar trend has been previously observed in the GC content of *Chlamydomonas eustigma*, an acidophilic green alga, in contrast to evolutionarily related neutrophilic green algae [81]. Consequently, low-GC Gram-positive bacteria of the phylum Firmicutes (*Desulfotomaculum*) are found in the upper size of the PCA, in contrast to a group of copiotrophic strains that are found in the bottom (Figure 3). Interestingly, this last group is composed by two subgroups. The first group is characterized by the predominance of lysogenic copiotrophs with middle-sized cells that contain a higher number of prophages and lower number of CRISPR arrays, and includes *D. aespoeensis* Aspo-2, *D. propionicus* type strain 1pr3, *D. fructosivorans* JJ, *D. alaskensis* G20, *D. africanus* DSM 2603 and *D. africanus* str. Walvis Bay (numbers 5, 21, 18, 44, 2, and 4 in Figure 3). The second subgroup is made up of lysogenic copiotrophs with smaller cells that contain a lower number of prophages and higher number of CRISPR arrays, including *D. vexinensis* DSM 17965, *Desulfovibrio* sp. A2, *D. desulfuricans* ND132, *D. putealis* DSM 16056, *D. elongatus* DSM 2908, and *D. oleovorans* Hxd3 (numbers 16, 29, 72, 57, 15, and 74 in Figure 3). The PCA also showed that there is a directly proportional relation between cellular volumes and number of prophages per cell (Spearman Correlation 0.3161, *p*-value = 0.03435) (Figure 3), indicating that bigger cells are more likely to contain a higher number of prophages. Conversely, the number of CRISPR per cell did not cluster closely with neither cellular volumes nor genome size. This contrast became more evident in the distribution of the proportion between SRP that contain prophages and CRISPR compared to those SRP that do not possess them (Figure 4). This finding suggests that there is an additional selective pressure for strains with smaller cells to get rid of additional “foreign” DNA, such as prophages, due to lower availability of cellular resources. Furthermore, there was no correlation between the number of prophages and number of CRISPR elements with the host genome size of lysogens (Spearman *p*-value = 0.9667 for prophages and *p*-value = 0.1906 for ncrispr).

The use of appropriate metrics in the quantification of prophages is required, since the amount of foreign DNA per cellular volume seems to be a more relevant measure than the amount of foreign DNA per length of the host genome. This observation is supported by the fact that the replication and maintenance of foreign DNA depends not only on the possible advantages that it provides, but also on the availability of permanent cellular resources. In addition, phenotypic and genotypic available data of SRP showed that different families have high variability in cell size, but do not have substantial differences in genome size. As an example, strains belonging to Peptococcaceae have bigger cell size (Median size ~3.53 μm^3^) compared to strains of other families (Median size ≤ 1.68 μm^3^), whereas the median of genome size of all the strains of SRPs belonging to the Desulfobacteraceae, Desulfohalobiaceae, Desulfovibrionaceae, Desulfobulbaceae, and Peptococcaceae families ranged between 3,916,410 and 3,969,010 bp (Appendix A, Appendix A). The disparities between cell volume and genome size of the host determined that Peptococcaceae strains registered one of the lowest values of DNA prophage per cell, but the highest prophage density (Appendix A). However, the interpretation of this finding and the utilization of different metrics should be critically revised to ensure they are environmental relevant. For instance, more sophisticated methods to improve the determination of the length of prophages are required, since our own method produces a rather arbitrary value obtained from the consensus between both detection tools (PhiSpy and PHASTER). Second, available phenotypic and genomic information on the SRP as well as bacteriophages is still scarce and could be family-specific biased, leading to misguided conclusions.

Strains that express flagella could increase the frequency of encounters with viral particles in aquatic environments, compared to those whose movement is restricted to diffusion and advective flows (Appendix A). This effect of flagella in phage-host relation has been portrayed by flagellotropic phages, a group of phages that infects *Agrobacterium*, *Asticcacaulis*, *Caulobacter,* and *Salmonella* strains, which are capable of sliding along the flagellum towards its site of insertion [82,83,84]. In *Caulobacter*, it has been hypothesized that those phages may even identify motile cells as an opportunity of increasing the likelihood of infection [85]. This does not seem to be the case of SRP, because no correlation between the presence of flagella and the number of prophages was observed (Fisher Test, *p*-value = 0.6399). While motility and growth rate are not relevant (Appendix A), the habitat of SRPs seems to be relevant for the presence of prophages. Indeed, 68% (15 out 22) of strains isolated or recovered exclusively from freshwater sediments are lysogens carrying a total of 22 prophages, whereas 43% (17 out 40) of strains isolated or recovered from either marine or brackish environments are lysogens (Appendix A). A similar trend was observed with CRISPR containing strains (Appendix A), and it is in agreement with previous reports that showed a higher production of viral particles in freshwater sediments than in coastal or sea sediments [86,87]. These results suggest that freshwater environments, which probably have a wider gradient of factors such as temperature, salinity, organic matter, may be a suitable habitat for a higher viral diversity [88].

Our results showed that the prevalence of CRISPR-Cas systems in SRPs is higher in thermophilic strains than in mesophilic strains, which have been reported in other microbial species [89]. All the thermophilic strains evaluated (10), except the archaea *Archaeoglobus profundus* DSM 563, contain CRISPR arrays. This prevalence is higher than the 78% (62 out of 70) of prevalence in mesophiles (Appendix A). More importantly, the median number of spacers in thermophilic SRP (47.5) was almost twice that of mesophilic SRPs (25.2), which is relevant since the number of spacers reflect the quantity of sequences stored in each CRISPR array and, therefore, their capability to provide protection [55].

### 3.3. SRP Prophages Are Mainly Myoviridae

In order to gain more insights about prophage–host interactions in the SRP community, the prophages were classified using VIRFAM, a program that search and classify head, neck and tail modules based on the Aclame database [40]. VIRFAM predicted that 59 out of 81 prophages encode for structural components that allowed their classification in the viral order Caudovirales. Prophages classified as Myoviridae were the most prevalent (44%; 36 out of 81), followed by Siphoviridae and Podoviridae with 23% and 6%, respectively (Appendix A). Myoviridae viruses are morphologically similar to T4-like phages, which are characterized by their rigid contractile tails with heads between 50 and 110 nm of diameter and genomes ranging between 100 and 300 kbp [10,90]. While Siphoviridae have longer, flexible and filamentous tails, Podoviridae phages contain very short tails. In general, both groups possess non-contractile tails, icosahedral or elongated heads of ~60 nm of diameter and smaller genomes (<100 kbp) [91]. Twenty-two prophages (28%) were assigned by VIRFAM as unclassified, and 6 of them were classified as small prophages (<30 kpb) (Appendix A).

This study indicates that the model strain *D. vulgaris* Hildenborough contain the highest number of prophages (6) among the SRP community. These results are in agreement with previous studies [31]. The distribution of prophages in this strain partially represents the distribution of prophages in the SRPs. All the prophages in *D. vulgaris* Hildenborough were classified in the order Caudovirales. While prophages 1R1, 1R2, 1R5, and 1R6 were classified as Myoviridae, 1R3 and 1R4 were classified as Siphoviridae and Podoviridae, respectively. In addition to *D. vulgaris* str. Hildenborough phages, other SRP phages have been previously isolated. This is the case of like-phages particles induced from pure cultures of *D. desulfuricans* subsp. *desulfuricans* str. ATCC 27,774 [92], and more recently, from *Desulfovibrio alaskensis* G20 [93]. Kazimura and Araki isolated a bacteriophage with a ~30 kbp genome from marine sediments capable of infecting *Desulfovibrio salexigens*, a salt-requiring SRP [25]. Furthermore, phages extracted from deep subsurface in hard rock aquifer were capable to infect *D. aespoeensis* Aspo-2, which was isolated from the same environment [26]. However, phages lost their ability to infect cells after one infection cycle, suggesting that surviving lysogens rapidly gained immunity against new infections. Interestingly, our method was capable to detect prophages in the genome of the five strains.

### 3.4. Desulfovibrio Prophages Exhibited High Degree of Homology

Functional analysis of the prophage proteins revealed that most of the SRP prophages contain genes encoding for recognizable phage-specific functions and life cycle of phages. These genes include genes encoding for functions related to the integration or excision of phage DNA at the bacterial chromosome, such as integrases, terminases, and nucleases, for which the 81 prophages contain 97, 69, and 107 proteins, respectively. Twenty-four out of 81 prophages do not contain integrases. The length of these prophages is lower than the length of prophages that contain at least one gene encoding for integrase (*p*-value = 0.0014 of Wilcoxon test), suggesting that integrases, as well as other essential phage genes, may have been lost after integration. In addition, many proteins associated with the viral structural proteins are present in the prophages, including head-associated (66), capsid-associated (44), tail-associated (266), and baseplate-associated (69) proteins. SRP prophages contain many genes that were annotated as hypothetical proteins or unknown functions, in agreement with previous studies. For example, the six prophages of *D. vulgaris* Hildenborough averaged 45% of the genes classified as hypothetical proteins or conserved hypothetical proteins, while the three prophages of *D. vulgaris* DP4 averaged 41% genes encoding hypothetical proteins. However, there are some prophages with higher percentage of genes with unknown function. This is the case of the prophages in *D. africanus* DSM 2603 and *D. vulgaris* Miyazaki, for which the genes with unknown functions covered a total of 57% and 54% of the genome, respectively (Appendix A).

In order to identify homologous regions among prophages that belong to the SRP community, a protein alignment based on DIAMOND was performed [39], in which each open reading frame (ORF) was compared to each other. The outcome of the pairwise comparison revealed that fifty-nine (65%) SRP prophages contain ≥10 homologous proteins, and 35 of those were prophages belonging to Desulfovibrionaceae family (Appendix A). Although all prophages found in the study displayed homology of at least a single protein (*e*-value 10^−4^ and alignment length > 20), a substantial set of proteins is unique across the prophage SRP community (Appendix A). Likewise, we found that 17 of those 22 unclassified prophages contain less than 10 homologs within other SRP prophages (Appendix A). In addition, the lack of homology with genes encoding structural viral proteins together with the fact that unclassified prophages were slightly smaller than classified prophages (~1000 bp of the median) suggests that those 17 prophages could be considered prophage remnants, satellite viruses or isolated phage genes, as a result of a permanent prophage domestication process as reported previously [6].

In contrast to homology, we observed a low level of organization of ORFs within the *Desulfovibrio* prophages that contain high level of homology (Appendix A). One exception is represented by the prophages 5R2 and 44R1 from *D. aespoeensis* Aspo-2 and *D. alaskensis* G20, respectively, which shows a high level of synteny throughout ~33 kbp (Figure 5 and Figure 6A). Furthermore, prophages 1R5 and 14R1, belonging to *D. vulgaris* Hildenborough and *D. vulgaris* str. Miyazaki, respectively, shared synteny along ~30 kbp, where genes encoding proteins involved in viral functions are predominant (Figure 5 and Figure 6B). This syntenic fragment is remotely interrupted in an intercalated manner by genes encoding hypothetical proteins and bacterial functions, suggesting that they may be the result of recent insertion events. Interestingly, pairwise comparison of the whole genomes of *D. vulgaris* Hildenborough and its close relative *D. vulgaris* DP4 found that both genomes were virtually identical. There are only 11 different regions of more than 20,000 bp between both genomes. Seven of those areas represent insertions of lysogenic prophages that were detected by our study, including two prophages found in *D. vulgaris* DP4, prophages 90R2 and 90R3, and all prophages found in *D. vulgaris* Hildenborough (Figure 7) highlighting the relevance of prophages and phages as a vector of genomic divergence in the SRP community.

### 3.5. Induction of Desulfovibrio Prophages

In this study, numerous assays to induce phages from SRP cultures were carried out. These attempts included induction of virus using mitomycin C in three strains of the *Desulfovibrio* genus, *D. vulgaris* str. Hildenborough, *D. africanus* 2603, and *D. acrylicus* DSM 10141. These strains were selected based on the presence or absence of prophages and CRISPR elements in their genomes, since *D. vulgaris* str. Hildenborough contain both prophages and CRISPR elements, *D. africanus* 2603 carry only prophages and *D. acrylicus* DSM 10,141 has neither prophages nor CRISPR systems. The induction was tested with two concentrations of mitomycin C (1 μg mL^−1^ and 3.5 μg mL^−1^) added at early exponential phase (Appendix A–C). The exposure to the two concentration of the antibiotic slightly reduced the growth rate of *D. vulgaris* str. Hildenborough. In contrast, the presence of the antibiotic did not affect the growth of *D. africanus* 2603 and *D. acrylicus* DSM 10141. Viral-like particles with similar morphotype of Podoviridae and structures that resembles outer membrane vesicles were observed in TEM micrographs of *D. vulgaris* str. Hildenborough after induction with mitomycin C (Appendix A). Membranes vesicles have been observed in *D. alaskensis* DSM16109, in cultures incubated in presence or absence of mitomycin C [93]. A previous study has suggested that after peptidoglycan lysis caused by endolysins, membrane vesicles can be formed from membranes [94]. Afterwards, in the present study we attempted to test cross infection among the three strains. The exposure to viral extracts obtained from *D. vulgaris* str. Hildenborough and *D. africanus* 2603 slightly reduced the growth rate of *D. africanus* 2603, only when the extract was added during exponential phase (Appendix A). Double-layer agar experiments based on the same treatments were performed. The development of plaques was not observed, however, the presence of iron sulfide precipitates may obscure the observation. Similar results were observed with normal and low iron concentration in the medium, highlighting that this limitation should be overcome to apply the techniques extensively used in phage research in SRP [95].

## 4. Conclusions

From an ecological perspective, the composition and dynamic of naturally occurring communities in pristine sulfidogenic environments are mainly shaped by two forces. The bottom up force selects for SRP that are either specialized for a limited number of substrates or capable to utilize a wide variety of nutrients, such as *Desulfovibrio piger* ATCC 29,098 or *D. desulfuricans* subsp. aestuarii ATCC 29578, respectively [96,97]. Thus, often the taxonomic and functional SRP diversity mirrors the diversity of substrates available. In contrast, the top down force refers to a predator–prey regulation, where viral predation plays an important role in reshaping microbial communities [26,90,98]. Since a limited number of SRP phages have been isolated and characterized, their role in the microbial ecology remained poorly understood. Our results demonstrate that there is a high prevalence of prophages and CRISPR elements, representing the 56% and 78% of the surveyed strains. Prophages and CRISPR arrays are non-uniformly distributed across the SRP phylogeny, where some lineages, such as Desulfovibrionaceae and Peptococcaceae, concentrated a higher number of prophages. We also found that there is a directly proportional relation between cellular volumes and number of prophages per cell, suggesting the opportunity cost of maintaining prophages, but not CRISPR, is less rewarding in cells with small volumes. These results are consistent with the fact that cellular volume and shape have been shown to be relevant for keeping nutrients diffusion efficiency required for free-living prokaryotes [99], especially those ones which habitats are not precisely profuse in energy sources. This is the case of SRP that face the permanent requirement of at least two moles of ATP in the activation of sulfate to APS in the first step for sulfate reduction, that burdens the cell’s energy budget [100]. From the phage perspective, ecophysiological attributes of the host are very important on adapting traits, such as adsorption rate, latent period, and burst size to be suitable for the environmental conditions in which both, virus and host, live. This study will contribute to improve the understanding of the abundance and distribution of prophages and CRISPR arrays and how they affect the microbial community dynamics in several sulfidogenic natural and engineered environments, including anoxic sediments, muds, stream and rivers, and sewage digesters.

## Figures and Tables

**Figure 1 microorganisms-09-00931-f001:**
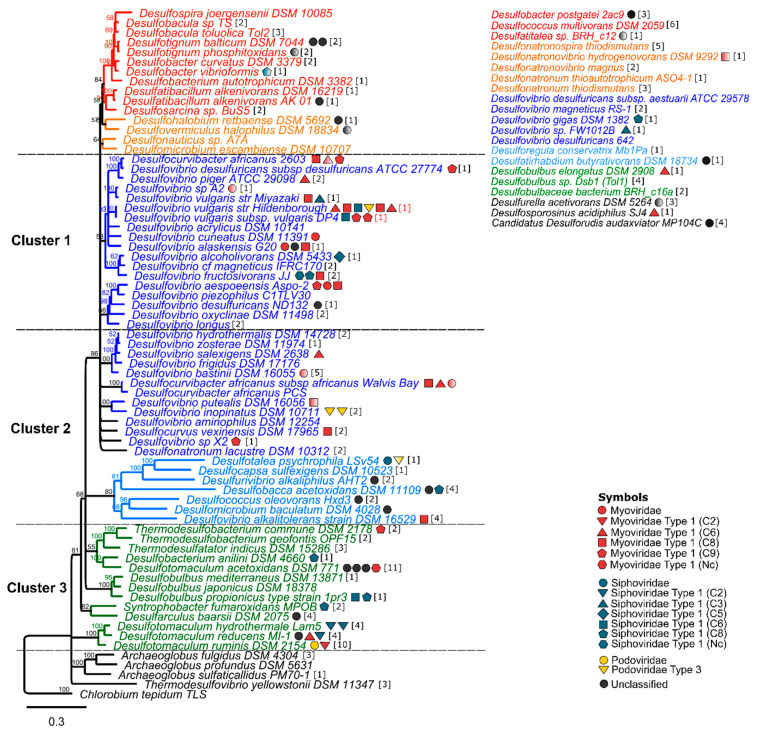
Distribution of prophages and CRISPR arrays found in SRP. The Neighbor-joining phylogeny of the DsrAB dissimilatory (bi)sulfite reductases sequences of 70 out of 91 SRP analyzed by this study. DsrAB dissimilatory (bi)sulfite reductases amino acid sequences were obtained from NCBI database and aligned using MUSCLE algorithm and manually curated in Geneious to a final alignment of 523 unambiguously aligned positions. The tree was rooted with the oxidative bacterial type DsrAB sequence obtained from *Chlorobium tepidum* as outgroup. Scale bar indicates 30% sequence divergence. Consensus support (%) is shown next to each node. Colored branches indicate families as the following. Red branches include strains belonging to Desulfobacteraceae; orange, include strains belonging to Desulfohalobiaceae, Desulfomicrobiaceae; blue, include strains belonging to Desulfovibrionaceae and Desulfonatronaceae; light blue, included strains belonging to Desulfobulbaceae, Syntrophacea, Desulfomicrobiaceae and Desulfovibrionaceae, green include strains belonging to Thermodesulfobiaceae, Desulfobacteraceae, Peptococcaceae, Desulfobulbaceae, Syntrophobacteraceae and Desulfarculaceae; and black branches include strains belonging to Archaeoglobaceae and Nitrospiraceae. Prophages are indicated after each lysogen by symbols and colors according to the Virfam, based on the classification of the ACLAME bacteriophages. Degrading colors indicate prophages were classified as small (<30 kb). The number of CRISPR elements found in each SRP is shown as a number in brackets. Twenty-one strains shown in the upper-right corner of the figure were excluded of the phylogeny due to either intact DsrAB sequences were not found in the genomes of those strains or presented high discrepancies with previous studies [56].

**Figure 2 microorganisms-09-00931-f002:**
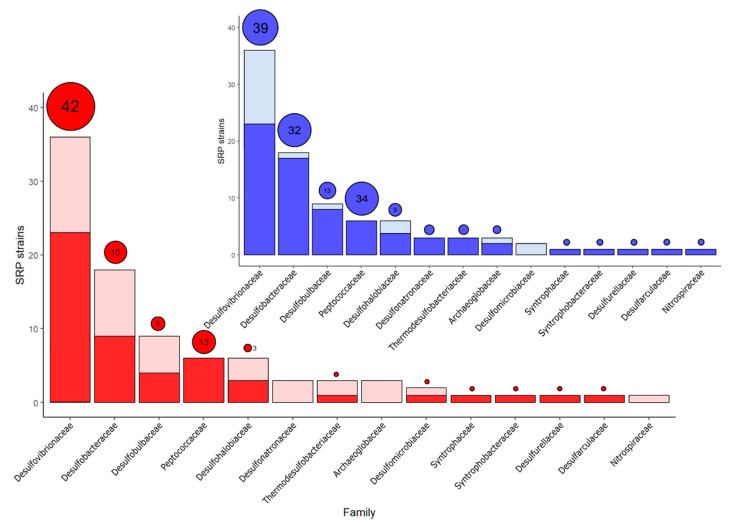
Distribution of SRP carrying prophages and CRISPR across different families. On the bottom chart, red and light red represent lysogenic and no lysogenic SRPs, respectively. Red circles represent the total number of prophages found in each family. On the top chart, blue and light blue represent CRISPR-containing and non CRISPR-containing SRP, respectively. Blue circles represent the total number of CRISPR arrays found in each family.

**Figure 3 microorganisms-09-00931-f003:**
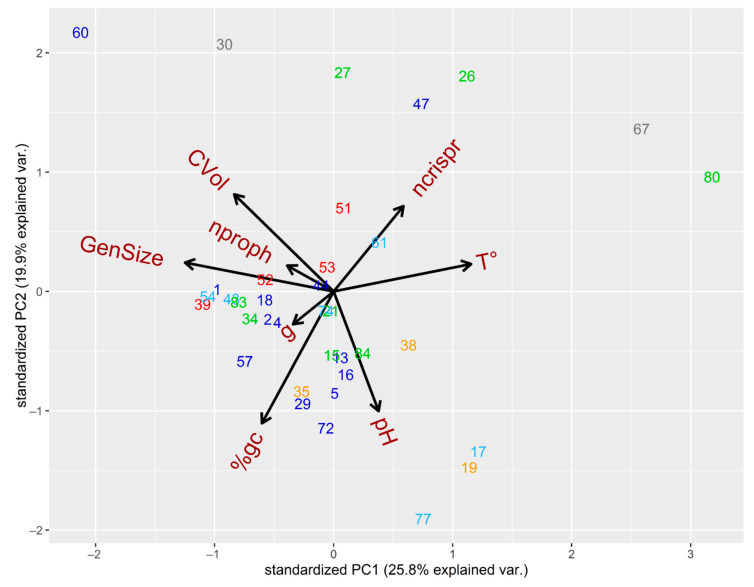
Principal component analysis of SRP according to their genomic and ecophysiological traits. Lysogenic SRP were compared based on the following eight metrics: number of prophages per cell (nproph), number of CRISPR (ncrispr), optimum pH for growth (pH), optimum temperature for growth (T°), GC content (% gc), minimum doubling time (g), cell volume (CVol), and genome size (GenSize). The numbers indicate de number of each SRP (according to Appendix A), and color indicates their phylogenetic affiliation (according to Figure 1). Strains *D. desulfuricans* subsp. *desulfuricans* str. ATCC 27,774 and *Desulfobulbus propionicus* type strain 1pr3 were excluded of the analysis, since they have the highest and lowest values of cellular volume.

**Figure 4 microorganisms-09-00931-f004:**
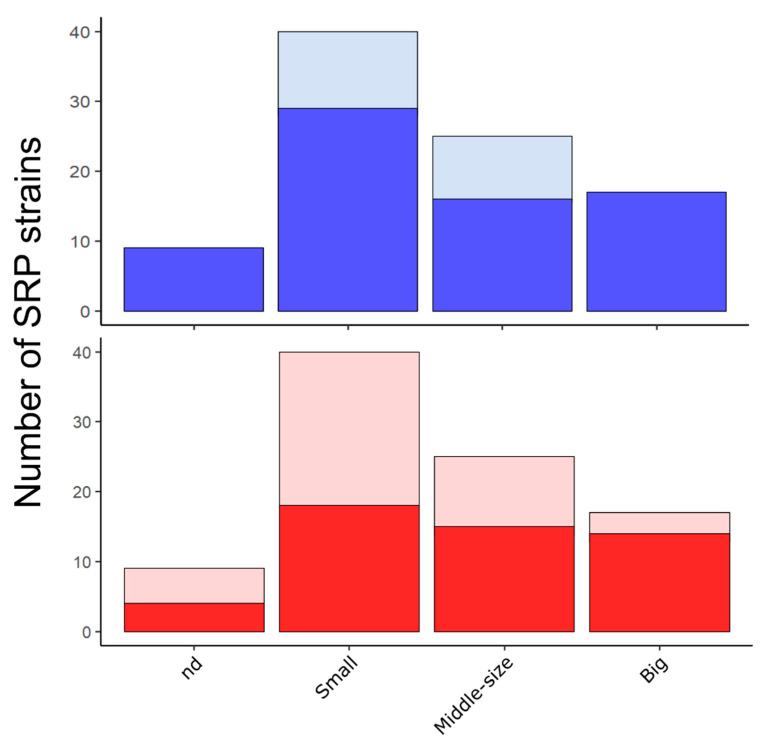
Distribution of CRISPR (**top**) and prophages (**bottom**) in different cell sizes. The top chart shows SRP that contain CRISPR (blue) and do not contain CRISPR (light blue). In the lower chart, lysogenic and non-lysogenic SRP are shown in red and light red, respectively.

**Figure 5 microorganisms-09-00931-f005:**
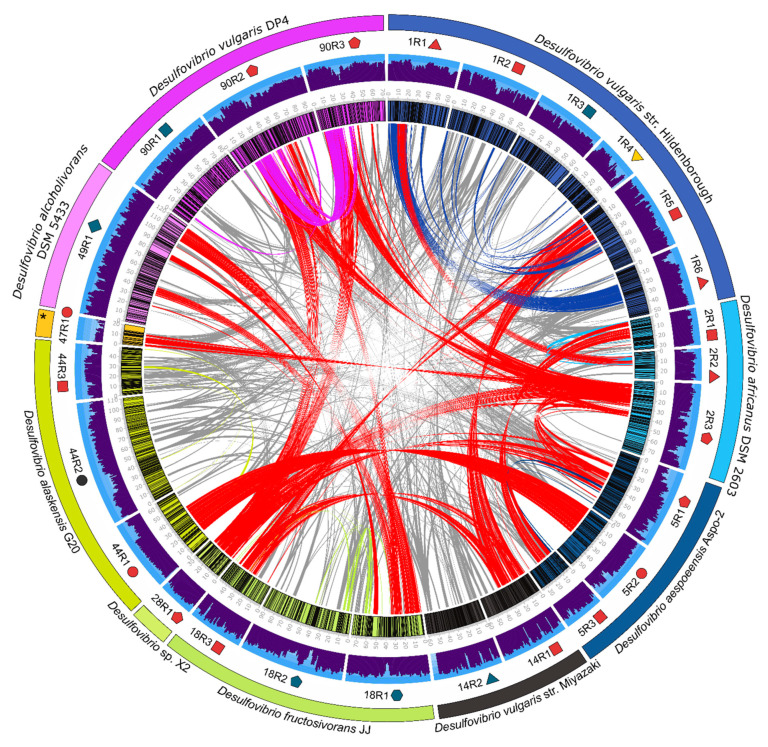
Circos plot shows the comparison of amino acid sequences among prophages of selected *Desulfovibrio* strains. Links show homologies between different prophage regions. Red connections indicate the most significant homologies between prophages from different hosts (more than 10 homologs), while gray links show prophages that share less than 10 homologs. Other colored connections show homologies between prophages from the same host. The symbols indicate the classification of each prophage according to VIRFAM (see Figure 1). The asterisk indicates the strain *Desulfovibrio bastinii* DSM 16055.

**Figure 6 microorganisms-09-00931-f006:**
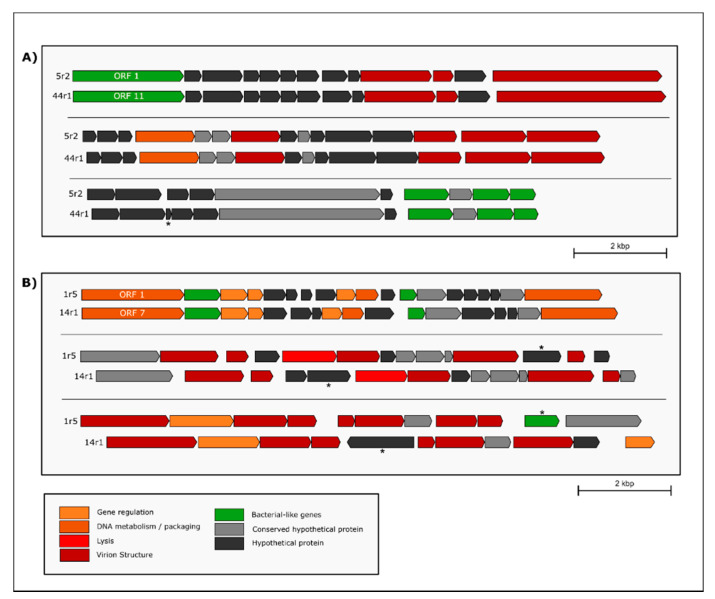
Comparative map of SRP prophages with a high level of synteny. (**A**) Fragment of high synteny between the prophages 5r2 (*Desulfovibrio aespoeensis* Aspo-2) and 44r1 (*Desulfovibrio alaskensis* G20), which contain a high prevalence of hypothetical proteins. (**B**) Fragment of high synteny between prophages 1r5 (*Desulfovibrio vulgaris* str. Hildenborough) and 14r1 (*Desulfovibrio vulgaris* str. Miyazaki). The asterisks indicate those ORFs not shared by both prophages. Annotation was made by Diamond and DNA Master.

**Figure 7 microorganisms-09-00931-f007:**
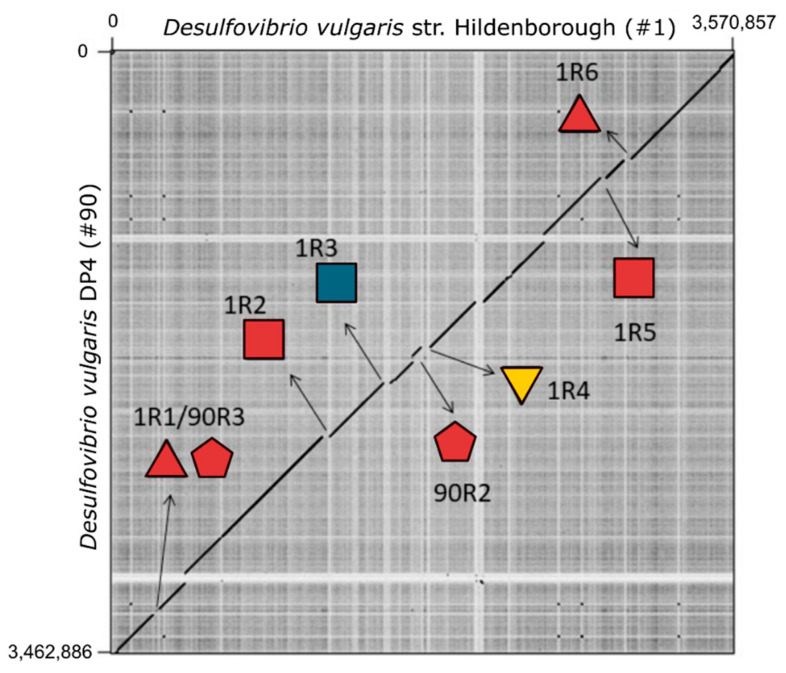
Dot plot matrix comparison calculated for the genomes of *D. vulgaris* Hildenborough and its close relative *D. vulgaris* DP4. Regions of similarity give rise to diagonal matches that are extended along a high proportion of the genomes. The *x*- and *y*-axis indicate full genomic chromosomal sequence comparisons of both SRP genomes. The arrows indicate seven regions that are associated with prophages found in both species. Prophages are indicated by symbols and colors according to the Virfam classification.

## Data Availability

The data presented in this study are openly available in https://github.com/robertoorellanar/Prophages_in_SRP.

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
