# Peer review of "Ecophysiological Features Shape the Distribution of Prophages and CRISPR in Sulfate Reducing Prokaryotes"

_microorganisms, 2021, doi:10.3390/microorganisms9050931_

Round 1

Reviewer 1 Report

This manuscript starts out talking about sulfate reducing bacteria and how prophages and CRISPR arrays of sulfate reducing bacteria have not been well enumerated. It then enumerates them. It then expands on the statement in the title that "ecophysiological features" shape the distributions of prophages and CRISPR's in sulfate reducing bacteria. The latter turns out to be a principal component analysis of various properties such as genome size, cell volume, numbers of prophages, and, unfortunately, nothing to do with sulfate metabolism, to see what collections of these properties tend to correlate among themselves. Their main conclusion is that cells with larger cell volume tend to have more prophages. This leads to a discussion about how "more resources", presumably represented by a larger cell volume, is an organizing feature with respect to carrying more prophages. Reading between the lines, I suspect they intend that sulfate metabolism is a contributor to these "resources". Also reading between the lines, it becomes clear at some point that the sulfur pathways have horizontally transferred, and so the authors may be hoping that prophages have something to do with moving these pathways about the microbial world.

There are a number of reasons to be interested in this manuscript. The authors have done an exceptional job of creating supplemental materials that allow the reader to look under the cover and ask further questions about what may be going on. For those purely interested in phageology, these are examples of taxa with few viruses characterized in the coat, but with lots of material in the form of prophage remnants that might be used to reconstruct a segment of the virosphere. That turns out to be hard to do, so someone attempting a fairly large scale effort such as this is welcome. For those interested in bacterial bioinformatics, the large fraction of genes known as "auxiliary" (those that reside in islands of horizontally transferred stuff) is an intellectual desert at this time. So demonstration of what might be accomplished with a different analytical method, in this case the PCA, is of interest. I personally have a soft spot for authors that try to do hard things, and so I hope that this work can make it to publication.

However, readers interested specifically in sulfate pathways are going to be disappointed at the lack of any obvious connections, and the overall fuzziness of the logic will be frustrating to others.

I spent some considerable time looking at issues in the manuscript. I'm going to confine my comments to a few points.

1) The authors enumerated prophages by using a couple of web sites to do it for them. I've had nothing but trouble with those sites, so I reworked the first 3 "prophages" in detail. 1R1 had an identifiable attachment site half way through the supposed "prophage". That integase uses a partial tRNA as attP encoded next to it, and reconstructs a functional tRNA as an attL or attR site. That tRNA next to the integrase is the end or the prophage. The rest of that 1R1 segment is not phage derived. 1R1 is missing the other att site and about one operon of early genes. The story for 1R2 and 1R3 is similar, although I didn't find either att site in those cases. But about 1/2 the genes labeled as "prophage" are not phage derived in that sample. I didn't work through all 91 "prophages", so these three may not be representative. But I have to tell you that this result is way better than I expected based on my prior experience with these web sites. The upshot is that the variable used in the PCA called phage length is nonsense. I can believe that the variable called number of prophages is a reasonable surrogate for the actual census of prophage remnants. The authors did acknowledge that by "prophage" they meant remnants of prophage, so that's OK. Using these automatons to count prophages is reasonable, since anything more exacting quickly gets pretty brutal. But the limitations have to be kept in sight. For example, enumerating the number of "hypothetical proteins" in these segments is doubly nonsensical. There is no distinction between what's prophage and what's other stuff, and whether it's "hypothetical protein" or something else mainly depends on how hard the annotator was willing to work (mostly commonly, not much).

2) Right from the top, by describing the sulfur reducing taxa only at the family level, the text obscured that these are a mixture of deltaproteobacteria and firmicutes (Gram positives). That immediately causes the reader to miss the implications about horizontal transfer inherent in Fig. 1. It also substantially obscures what's going on in the PCA.

3) My understanding of how to interpret a PCA is to independently consider how each vector loads on PCA1 and PCA2. Concomitantly, you must model what underlying principles might be causing a significant portion of variation in multiple parameters to coordinate on those two axes. In this case, PC1 clearly is coordinated by the oligotrophy theory, whereby small genome bacteria are specialists and typically deal with nutrient poor or other difficult environments, and large genome bacteria are generalists and typically attempt to dominate nutrient rich environments. I didn't immediately recognize what PC2 was here until I started decoding the identities of the individual bacteria. PC2 is the taxonomic axis with Firmicutes (low %GC) up and Gram negatives down. All the systematic differences between Gram positives and deltaproteobacteria are going to correlate along that axis. For example, consider %GC. What's known about %GC is that it correlates a little bit with genome size within phylum, but much more to systematic shifts between phyla. So, it loads heavily on PC2, and a little bit on PC1, exactly as it should. One thing I like about this manuscript is that shows me an excellent way to present exactly that relationship, whereas every effort in the literature to present that relationship so far is ugly and obtuse.

The manuscript didn't interpret its results consistent with PC1 and PC2 representing two different underlying principles. The focus on the cell volume and number of prophages vectors apparently originated because they point in the same diagonal direction. That diagonal was treated that like a principal component axis. It's not a principal component axis (an axis that explains lots of variation in multiple parameters and hence is probably not a coincidence). Hence there is no reason to think the coordination of those two vectors is any more than coincidence. Keep in mind that every vector has to point somewhere, so as more vectors are loaded onto the plot, there will be coincidences. The analysis can be extended to PC3, PC4, PC5..., but that will just produce progressively smaller numbers for percent of total variation explained, which is another way of saying more likely to be a coincidence.

So I'd interpret the cell volume result as so: on PC1, cells with larger genomes make larger cells, and independently on PC2, cells with cell walls armored against osmotic shock tend to be larger cells. The reasons underlying loading on PC1 and PC2 are supposed to be independent, moving the same set of parameters in different patterns for two different underlying reasons.

Similarly, for numbers of prophages, cells with smaller genomes have fewer prophages; that doesn't seem to take much imagination. It isn't terribly important to say small genome versus small cell volume versus living in cold austere environments. All those things are correlated as a single thing: variously called oligotrophs or specialists. I would emphasize the genome length because conceptually quiescent prophages don't affect anything but genome length, and that's where the genetic action to allow them or restrict them is going to take place. Independently, Gram positives have more prophages than deltaproteobacteria. That's interesting. One might then think of an interesting hypothesis and try to test it. Or one might find out that it's an ascertainment bias, since the prophages are being counted by similarity to known phages, and there are a lot more known phages in Firmicutes to be similar to than there are in deltaproteobacteria.

The manuscript reports that there are not correlations of CRISPRs and prophages. On PC1 smaller genomes have more CRISPRs and fewer prophages. This invites the attractive hypothesis that CRISPRs are part of the mechanism that small genome bacteria use to prevent prophages from blowing up their genomes. The manuscript says that by a Spearman statistic, the CRISPR result is not significant but the prophage result is significant. But the CRISPR load on PC1 is higher than the nprophage load, so I'm confused about how it could be less significant. Doesn't the Spearman statistic evaluate monotonicity? I'm not sure I would expect monotonicity on PC1 after the combined action with PC2.

Independently, there are more CRISPRS and more prophages in Gram positives by comparison to deltaproteobacteria. Again, there might be something interesting to track down there. But one has to again consider the possibility of an ascertainment bias, since CRISPRs are also counted by similarity to know sequences, and everything about Firmicutes is based on a much larger knowledge base than deltaproteobacteria. Note that smaller genomes having more CRISPRs can be true in both deltaproteobacteria and Firmicutes even while Firmicutes has more CRISPRs and more prophages than deltaproteobacteria. The authors should probably test that if there are enough Gram positives in the sample.

I confess I keep getting turned around by the directionality of the temp vector. Are there thermophilic small genome bacteria in the sample; am I reading that right? A table in supplementary material of the numbers underlying the PCA figure would be helpful.

Let me emphasize that I'm not a statistician and, consistent with the 1 week review policy, I'm writing without having worked it over enough to know what I'd conclude if it was my own work. So, I can not insist that the authors must agree with my interpretations. I am just giving what feedback I can.

Minor criticism:

The legend to fig. 3 says the color code is as on fig.1, but there is no orange on fig. 1.

Reviewer 2 Report

In this paper, the authors explore the drivers of virus-host interactions in sulfate reducing prokaryotes (SRPs), which are critical components of the sulfur cycle. The authors investigate the occurrence and distribution of prophages and CRISPR arrays in published genomes of SRPs, evidencing a non-uniform distribution of prophages among the cellular taxa analyzed and a high prevalence of CRISPR arrays. Furthermore, the authors curate a thorough compendium of ecological and physiological traits and investigate if these features influence the distribution of prophages, a premise seldomly addressed and thus interesting in the field of virus-host interactions.

The introduction gives a satisfactory overview of the literature, the methods employed to predict prophages and CRISPR arrays are solid, and the list of ecophysiological traits in the SRPs analyzed is carefully done. However, I have some comments about the results obtained in this work.

Major comments:

  1. While the paper addresses the influence of ecophysiological traits in SRP-prophage distribution, it is unclear from the results which features, beyond cell volume, affect prophage occurrence in SRPs. Perhaps the title should be modified to better reflect the results of the paper.
  2. Figure 1. This figure presents a phylogeny based on the DsrAB enzyme that may not reflect the organism taxonomy, but agrees with the authors purpose of relating physiological traits to prophage and CRISPR array prevalence in SRPs. I have three major comments here. First, the presence of organisms in the upper-right corner that are not part of the phylogenetic tree is not explained in the text or legend. Second, the rationale behind the classification of SRPs into clusters is not mentioned in the text and I believe the understanding of the paper would be greatly improved by explaining this point, given that I infer this classification is based in ecophysiological properties. Lastly, it seems to me that this figure refers to two different clustering systems, the one mentioned above referring to the SRPs and a second, putatively referring to the prophages Virfam classification. Given that both have the same notation, it complicates the understanding of the figure and I suggest the authors to rename one of them.
  3. L 313-387 appear to me one of the main attempts to link ecophysiology to prophage distribution. Three clusters are defined by the authors, but the rationale behind this classification needs to be addressed in Methods or in the text. While cluster 1 and 2 appear to have common features, the traits linking members of cluster 3 are less clear and this group seems very heterogeneous. Furthermore, it appears that microorganisms from closely related taxons have quite different ecophysiology (e.g. Desulfovibrio members being divided into two clusters). A better explanation of this classification would greatly aid the understanding of the paper.
  4. While the authors argue that the ecophysiological clusters above mentioned influence prophage distribution, virus taxonomy does not appear to correlate with host ecophysiology. Perhaps a more canonical taxonomic classification of the microorganisms would better reflect the occurrence of prophage taxons, as suggested by Fig. 5, where the most closely related viral genomes were all present in the genus Desulfovibrio.
  5. Figure 4. This PCA of ecophysiological traits supports a correlation between cell volume and number of prophages that the authors find statistically significant, which I find a very interesting point. However, their analysis does not support other relationships between prophage occurrence and ecophysiological traits. The authors also point that CRISPR array numbers do not correlate with any variable tested. This undermines the authors´ premise that ecophysiology shapes prophage and CRISPR array distribution in SRPs. 
  6. Section 3.3. I believe this section can be further summarized, especially the description about phages induced from vulgaris str. Hildeborough, since it is not directly linked to the authors´ results.
  7. In my opinion, this section is better described as the conclusions of the paper, while the section of Results is a combination of both results and discussion. I suggest the authors to adjust the organization of the manuscript.

Minor comments:

i) L 15-17. “Although their environmental relevance, …” -> change to “Despite their environmental relevance,...”

ii) L 233. The phrase “…, they use should be appropriated” needs to be revised.    

iii) L 372-3737. It is suggested to change “partner” to “partners” and improve clarity about how those organisms relate to each other.

iv) L 403. Change “CRIPSR” to “CRISPR”

v) Section 3.1. While the authors did not find a correlation between the presence of CRISPR and the presence of prophages, they point that the prevalence of CRISPR in SRPs is higher than the general prevalence of CRISPR in other Bacteria, and that perhaps this reflects an increased pressure to get rid of prophages than CRISPR. One important source of selective pressure is the presence of CRISPR spacers that target the prophage genome, and I agree when the authors write that a more detailed characterization of the CRISPR diversity may shed light over these virus-host dynamics.

vi) Replace “prophage” for “CRISPR” in the legend of the x axis of Fig. S5F.

vii) Figures S8 and S9. The authors relate prophages by protein homology, but the use of hosts as nodes when a host may harbor prophages of different viral taxons may obscure the relationships between prophages if more than one virus is present in the genome. Which I think is what happens in this figure, where the similitude between prophages is hard to discern. Figure 5 is a better summary of the authors results regarding the right panel of Fig. S8, in my opinion. While this analysis was done based on protein homology, a nucleotide comparison was used to evaluate synteny in the prophage genome. Viruses are known to vary greatly at nucleotide level, so it is understandable that only a couple of viruses showed synteny in Fig S9. Perhaps an evaluation of synteny at protein level would have given better results.

Round 2

Reviewer 1 Report

A reasonable effort to meet the criticisms has been made.  I have no further comments.

Reviewer 2 Report

Orellana et al. have addressed my comments satisfactorily and I would like to recommend the publication of this version of the manuscript.